# Attacking Graph Classification via Bayesian Optimisation

Xingchen Wan [1]  Henry Kenlay [1]  Binxin Ru [1]  Arno Blaas [1]  Michael A. Osborne [1]  Xiaowen Dong [1]

## Abstract

Graph neural networks have been shown to be vulnerable to adversarial attacks. While the majority of the literature focuses on such vulnerability in node-level classification tasks, little effort has been dedicated to attacks on graph-level classification, an important problem with numerous real-life applications such as biochemistry and social network analysis. The few existing methods often require unrealistic setups, such as access to internal information of the victim models, or an impractically-large number of queries. We present a novel Bayesian optimisation-based attack method for graph classification models. Our method is *black-box*, *query-efficient* and *parsimonious* with respect to the perturbation applied. We empirically validate the effectiveness and flexibility of the proposed method and analyse patterns behind the adversarial samples produced, which may shed further light on the adversarial robustness of graph classification models.

## 1. Introduction

Graphs are a general-purpose data structure consisting of entities represented by nodes and edges which encode pairwise relationships. In recent years, graph-based machine learning models has been widely used in a variety of important applications such as semi-supervised learning, link prediction, community detection and graph classification (Cai et al., 2018; Hamilton, 2020; Zhou et al., 2020).

Despite the growing interest, it has been shown that, like many other machine learning models, graph-based models are vulnerable to adversarial attacks (Jin et al., 2020; Sun et al., 2018). Adversarial attacks on graphs can be aimed at different learning tasks. This paper focuses on graph-level classification, where given an input graph, we wish to

learn a function that predicts a property of interest related to the graph. Graph classification is an important task with many real-life applications, especially in bioinformatics and chemistry (Morris et al., 2020a;b). For example, the task may be to accurately classify if a molecule, modelled as a graph whereby nodes represent atoms and edges model bonds, inhibits HIV replication or not. Although there are a few attempts on performing adversarial attacks on graph classification (Dai et al., 2018; Ma et al., 2019), they operate under unrealistic assumptions such as the need to query the target model a large number of times or access a portion of the test set to train the attacking agent.

To address these limitations, we formulate the adversarial attack on graph classification as a black-box optimisation problem and solve it with Bayesian optimisation (BO), a query-efficient state-of-the-art black-box optimiser. Unlike existing work, our method is query efficient, parsimonious in perturbations and does not require supervised training on a labelled dataset to effectively attack a new sample. Another benefit of our method is that it can be easily adapted to perform various modes of attacks such as deleting or rewiring edges and node injection. Furthermore, we investigate the topological properties of the successful adversarial examples found by our method and offer valuable insights on the connection between the graph topology change and the model robustness.

## 2. Proposed Method: GRABNEL

**Problem Setup**    A graph $\mathcal{G} = (\mathcal{V}, \mathcal{E})$ is defined by a set of nodes $\mathcal{V} = \{v_i\}_{i=1}^n$ and edges $\mathcal{E} = \{\mathbf{e}_i\}_{i=1}^m$ where each edge $\mathbf{e}_k = \{v_i, v_j\}$ connects between nodes $v_i$ and $v_j$. The overall topology can be represented by the adjacency matrix $\mathbf{A} \in \{0, 1\}^{n \times n}$ where $\mathbf{A}_{ij} = 1$[1] if the edge $\{v_i, v_j\}$ is present. The attack objective in our case is to degrade the predictive performance of the pretrained victim graph classifier $f_\theta$ by finding a graph $\mathcal{G}'$ perturbed from the original test graph $\mathcal{G}$ (ideally with the minimum amount of perturbation) such that $f_\theta$ produces an incorrect class label for $\mathcal{G}$. In this paper, we consider the *black-box evasion* attack setting, where the adversary agent cannot access or modify the the victim model $f_\theta$ (i.e. its network architecture, its

---

[1]Machine Learning Research Group, University of Oxford, Oxford, UK. Correspondence to: Xingchen Wan <xwan@robots.ox.ac.uk>.

*Accepted by the ICML 2021 workshop on A Blessing in Disguise: The Prospects and Perils of Adversarial Machine Learning.* Copyright 2021 by the author(s).

---

[1]We discuss the unweighted graphs for simplicity; our method may also handle other graph types.

**Algorithm 1** Overall pseudocode of the GRABNEL routine.

1: **Input:** Original graph $\mathcal{G}_0$, victim model $f_\theta$, $n_{\text{init}}$ (the number of random initialising points), Query budget $B$, Perturbation budget $\Delta$.
2: **Output:** An adversarial graph $\mathcal{G}^*$
3: Set base graph $\mathcal{G}_{\text{base}} \leftarrow \mathcal{G}_0$; initialise stage count `stage` $\leftarrow 0$.
4: Randomly sample $n_{\text{init}}$ perturbed graphs $\{\mathcal{G}'\}_{i=1}^{n_{\text{init}}}$ that are 1 edit distance different from $\mathcal{G}$ and query each perturbed graph to obtain their attack losses $\mathcal{L}_{\text{attack}}(f_\theta, \mathcal{G}'_i)$.
5: Compute the WL feature encoding for all graphs: $(\Phi(\mathcal{G}'_1), \ldots, \Phi(\mathcal{G}'_{n_{\text{init}}})) = $ `WLFeatureExtract`$(\mathcal{G}_0, (\mathcal{G}'_1, \ldots \mathcal{G}'_{n_{\text{init}}}))$. *// See App.**B** for details of* `WLFeatureExtract`*.*
6: Fit the sparse Bayesian linear regression surrogate with the data $\{\Phi(\mathcal{G}'_i), \mathcal{L}_{\text{attack}}(f_\theta, \mathcal{G}'_i)\}_{i=1}^{n_{\text{init}}}$
7: Divide total budget of $B$ into $\Delta$ stages *// See "Sequential perturbation selection"*
8: **while** query budget is not exhausted and attack has not succeeded **do**
9:    **if** query budget of the *current stage* is exhausted **then**
10:       Increment the stage count `stage` $\leftarrow$ `stage` $+ 1$ and update the base graph $\mathcal{G}_{\text{base}}$ with the graph leading to largest increase in attack loss in the previous stage. *// Refer to Fig. 1*
11:    **end if**
12:    Propose graph to be queried next $\mathcal{G}'_{\text{proposal}}$ via acquisition optimisation. *// See "Optimisation of acquisition function" in App. **B***
13:    Query $f_\theta$ for the graph proposed in the previous step to calculate its attack loss.
14:    **if** attack succeeded **then**
15:       Set $\mathcal{G}^* \leftarrow \mathcal{G}'_{\text{proposal}}$ and **return** it.
16:    **end if**
17:    Augment the observed data: $\mathcal{D} \leftarrow \mathcal{D} \cup \{\mathcal{G}'_{\text{proposal}}, \mathcal{L}_{\text{attack}}(f_\theta, \mathcal{G}'_{\text{proposal}})\}$, update the WL feature encodings of *all* observed graphs $(\Phi(\mathcal{G}'_1), \ldots, \Phi(\mathcal{G}'_{|\mathcal{D}|})) = $ `WLFeatureExtract`$(\mathcal{G}_0, (\mathcal{G}'_1, \ldots \mathcal{G}'_{|\mathcal{D}|}))$ and re-fit the surrogate.
18: **end while**
19: **return** None *// Failed attack within the query budget*

weights $\theta$ or gradients) or its training data $\{(\mathcal{G}_i, y_i)\}_{i=1}^L$; the adversary can only interact with $f_\theta$ by querying it with an input graph $\mathcal{G}'$ and observe the model output as pseudo-probabilities over all classes $f_\theta(\mathcal{G}') \in [0, 1]^C$. Additionally, we assume that *sample efficiency* is highly valued; the number of queries should be as few as possible to avoid detection in a real-life scenario. Formally, the objective can be formulated as a black-box maximisation problem on graph $\mathcal{G}$ as the objective function of our BO attack agent:

$$\max_{\mathcal{G}' \in \Psi(\mathcal{G})} \mathcal{L}_{\text{attack}}\big(f_\theta(\mathcal{G}'), y\big) \ \text{s.t.} \ y = \arg\max f_\theta(\mathcal{G}) \quad (1)$$

where $f_\theta$ is the pretrained victim model that remains fixed in the evasion attack setup and $y$ is the correct label of the original input $\mathcal{G}$. Denote the output logit for the class $y$ as $f_\theta(\mathcal{G})_y$, the *attack* loss $\mathcal{L}_{\text{attack}}\big(f_\theta(\mathcal{G}'), y\big)$ is given by $\max_{t \in \mathcal{Y}, t \neq y} \log f_\theta(\mathcal{G}')_t - \log f_\theta(\mathcal{G}')_y$ for *untargeted attack*, or $\log f_\theta(\mathcal{G}')_t - \log f_\theta(\mathcal{G}')_y$ for *targeted attack* on class $t$, where $f_\theta(\cdot)_t$ denotes the logit output for class $t$. Such attack loss definition is commonly used both in the traditional image attack and the graph attack literature (Carlini and Wagner, 2017; Zügner et al., 2018). Furthermore, $\Psi(\mathcal{G})$ refers to the set of possible $\mathcal{G}'$ generated from perturbing $\mathcal{G}$. In this work, we experiment with a diverse modes of attacks to show that our attack method can be generalised to different set-ups including but not limited to creating/removing edges and rewiring/swapping edges. The overall routine of GRABNEL is presented in Algorithm 1, and we now elaborate each of its key components, and we describe the method to optimise acquisition function in App. A

**Surrogate model** we propose to first use a *Weisfeiler-Lehman (WL) feature extractor* to extracts a vector space representation of $\mathcal{G}$, followed by a *sparse Bayesian linear regression* which balances performance with efficiency and gives an probabilistic output. With reference to Algorithm 1, given a perturbation graph $\mathcal{G}'$ as a proposed adversarial sample, the WL feature extractor first extracts a vector representation $\phi(\mathcal{G}')$ in line with the WL subtree kernel procedure (but without the final kernel computation) (Shervashidze et al., 2011). For the case where the node features are discrete, let $x^0(v)$ be the initial node feature of node $v \in \mathcal{V}$, we iteratively aggregate and hash the features of $v$ with its neighbours, $\{u_i\}_{i=1}^{\deg(v)}$, using the original WL procedure at all nodes to transform them into discrete labels:

$$x^{h+1}(v) = \text{hash}\Big(x^h(v), x^h(u_1), \ldots, x^h(u_{\deg(v)})\Big), \quad (2)$$

for all $h \in \{0, 1, \ldots, H-1\}$ and $H$ is the total number of WL iterations, a hyperparameter of the procedure. At each level $h$, we compute the feature vector $\phi_h(\mathcal{G}') = [c(\mathcal{G}', \mathcal{X}_{h1}), \ldots, c(\mathcal{G}', \mathcal{X}_{h|\mathcal{X}_h|})]^\top$, where $\mathcal{X}_h$ is the set of distinct node features $x^h$ that occur in all input graphs at the current level and $c(G', x)$ is the counting function that counts the number of times a particular node feature $x$ appears in $G'$. For the case with continuous node features and/or weighted edges, we instead use the modified WL procedure proposed in Togninalli et al. (2019):

$$x^{h+1}(v) = \frac{1}{2}\Big(x^h(v) + \frac{1}{\deg(v)} \sum_{i=1}^{\deg(v)} w(v, u_i) x^h(u_i)\Big), \quad (3)$$

for all $h \in \{0, 1, \ldots, H-1\}$, and we simply have $\phi_h(\mathcal{G}') = \text{vec}(X_h)$ where we vectorise the feature matrix of graph $\mathcal{G}'$ at level $h$. In both cases, at the end of $H$ WL iterations we obtain the feature vector $\phi(\mathcal{G}') = \text{concat}\Big(\phi_1(\mathcal{G}'), \ldots, \phi_H(\mathcal{G}')\Big)$ for each training graph in $[1, n_{\mathcal{G}'}]$ to form the feature matrix $\Phi = [\phi(\mathcal{G}'_1), \ldots \phi(\mathcal{G}'_{|n_{\mathcal{G}'}|})]^\top$ to be passed to the Bayesian regressor. The WL iterations capture both information related to individual nodes and topological information (via neighbourhood aggregation), and have been shown to have comparable distinguishing power to some GNN models (Morris et al., 2019), and hence the procedure is expressive. At the same time, the extraction $\mathcal{G}' \to \phi(\mathcal{G}')$ is also *unsupervised*, thereby avoiding the need for the surrogate to learn representation from the data to ensure good sample efficiency.

When $H$ or $\mathcal{G}'$ (with many WL features) are large, the resulting feature matrix will likely be very high-dimensional, which would lead to high-variance regression coefficients $\alpha$ being estimated if the number of input samples is comparatively few. To attain a good predictive performance in such a case, we employ Bayesian regression surrogate with the automatic relevance determination (ARD) prior to learn

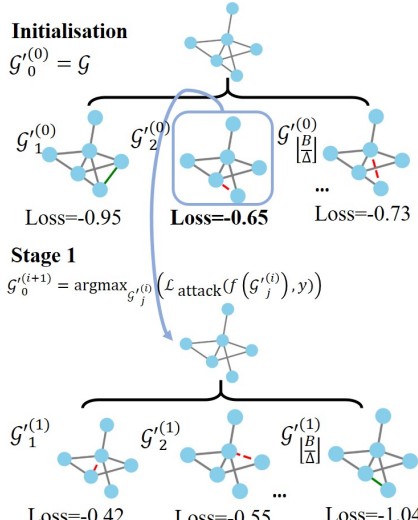

**Initialisation**
$\mathcal{G}'^{(0)}_0 = \mathcal{G}$

$\mathcal{G}'^{(0)}_1$  $\mathcal{G}'^{(0)}_2$  $\mathcal{G}'^{(0)}_{\lfloor\frac{B}{\Delta}\rfloor}$

Loss=-0.95  **Loss=-0.65**  Loss=-0.73

**Stage 1**
$\mathcal{G}'^{(i+1)}_0 = \text{argmax}_{\mathcal{G}'^{(i)}_j}\left(\mathcal{L}_{\text{attack}}(f\left(\mathcal{G}'^{(i)}_j\right), y)\right)$

$\mathcal{G}'^{(1)}_1$  $\mathcal{G}'^{(1)}_2$  $\mathcal{G}'^{(1)}_{\lfloor\frac{B}{\Delta}\rfloor}$

Loss=-0.42  Loss=-0.55  Loss=-1.04

*Figure 1.* Sequential edge selection. At each stage the BO agent sequentially proposes candidate graphs with edge edit distance of 1 from the base graph $\mathcal{G}'^{(i)}_0$ (which is the original unperturbed graph $\mathcal{G}$ at initialisation, or a perturbed graph that led to the largest increase in loss from the previous stage otherwise). This procedure repeats until either the attack succeeds (i.e. we find a graph $\mathcal{G}'$ with $\mathcal{L}_{\text{attack}}(f(\mathcal{G}'), y) > 0$) or the maximum number of $B$ queries to $f_\theta$ is exhausted.

the mapping $\mathbf{\Phi} \rightarrow \mathcal{L}_{\text{attack}}(f_\theta(\mathcal{G}'), y)$, which regularises weights and encourages sparsity in $\boldsymbol{\alpha}$ (Wipf et al., 2007):

$$\mathcal{L}_{\text{attack}}|\mathbf{\Phi}, \boldsymbol{\alpha}, \sigma_n^2 \sim \mathcal{N}(\boldsymbol{\alpha}^\top\mathbf{\Phi}, \sigma_n^2\boldsymbol{I}),$$
$$\boldsymbol{\alpha}|\boldsymbol{\lambda} \sim \mathcal{N}(\mathbf{0}, \mathbf{\Lambda}),\ \text{diag}(\mathbf{\Lambda}) = \boldsymbol{\lambda}^{-1} = \{\lambda_1^{-1}, ..., \lambda_{\dim(\boldsymbol{\lambda})}^{-1}\},$$
$$\lambda_i \sim \text{Gamma}(k, \theta)\ \forall i \in [1, \dim(\boldsymbol{\lambda})], \tag{4}$$

where $\mathbf{\Lambda}$ is the diagonal covariance matrix. To estimate $\boldsymbol{\alpha}$ and the noise variance $\sigma_n^2$, we optimise the model marginal log-likelihood. Overall, the WL routines scales as $\mathcal{O}(Hm)$, whereas training of Bayesian linear regression has a linear scaling w.r.t. the number of queries to the victim model; these ensure the surrogate is scalable to both larger graphs and/or a large number of graphs, both of which are commonly encountered in graph classification. Other options include GP and Bayesian neural network surrogates. While theoretically more expressive, empirically we find they perform similarly in this particular case, but are more noticeably more expensive than linear models.

**Sequential perturbation selection**  In the default structural perturbation setting, given an attack budget of $\Delta$ (i.e. we are allowed to flip up to $\Delta$ edges from $\mathcal{G}$), finding exactly the set of perturbations $\delta\mathbf{A}$ that leads to the largest increase in $\mathcal{L}_{\text{attack}}$ entails an combinatorial optimisation over $\binom{n^2}{\Delta}$ candidates. This is a huge search space that is difficult for the surrogate to learn meaningful patterns in a sample-efficient way even for modestly-sized graphs. To tackle this challenge, we adopt the strategy illustrated in Fig.

1: given the query budget $B$ (i.e. the total number of times we are allowed to query $f_\theta$ for a given $\mathcal{G}$), we amortise $B$ into $\Delta$ stages and focus on selecting *one* edge perturbation at each stage. While this strategy is greedy in the sense that it always commits the perturbation leading to the largest increase in loss at each stage, it is worth noting that we do *not* treat the previously modified edges differently, and the agent can, and does occasionally as we observe empirically, "correct" previous modifications by flipping edges back: this is possible due to the edge selection being permutation invariant. Another benefit of this strategy is that it can potentially make full use of the entire attack budget $\Delta$ *while* remaining parsimonious w.r.t. the amount of perturbation introduced, as it only progresses to the next stage and modifies the $\mathcal{G}$ further when it fails to find a successful adversarial example in the current stage.

## 3. Related Works

Recently there has been an increasing attention in the study of adversarial attacks in the context of graph neural networks (Jin et al., 2020; Sun et al., 2018). Applying adversarial attack methods from other domains to the graph setting is not straightforward, especially those based on gradient information, since the graph domain is inherently discrete. One of the earliest models, Nettack, attacks a GCN node classifier by optimising the attack loss of a surrogate model using a greedy algorithm (Zügner et al., 2018). DICE attacks node classifiers by adding edges between nodes of different classes and deleting edges connecting nodes of the same class (Waniek et al., 2018). Several reinforcement learning-based techniques (Ma et al., 2019; Zügner et al., 2018) have also been introduced for attacks on both node and graph classifiers . Compared to both these methods, our method does not use the training data of the victim model and is much more query efficient. Adversarial attacks on graph classifiers outside of the evasion setting have also been considered such as backdoor attacks (Xu et al., 2021; Zhang et al., 2020). On the other hand, BO as a black-box evasion attacking agent has been explored on tabular (Suya et al., 2017) and image data (Munoz-González, 2017; Ru et al., 2019; Shukla et al., 2019; Zhao et al., 2019). However, we address the problem for graph classification models, which work on structurally and topologically fundamentally different inputs, implying several nontrivial challenges that require our method to go beyond the vanilla usage of BO.

## 4. Experiments

In this section, we validate the performance of the proposed method in a number of graph classification tasks. All additional details, including the statistics of the datasets used and implementation details of the victim models and attack methods, are presented in App. D.

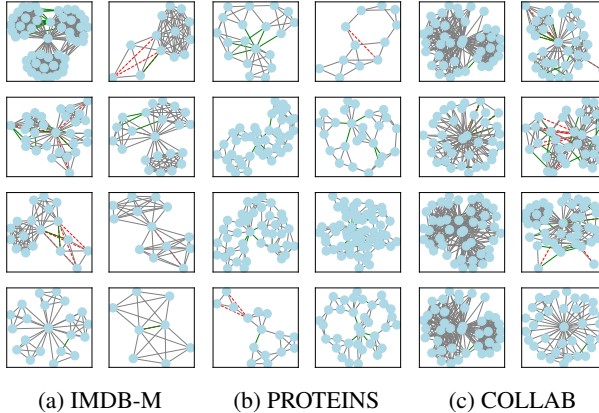

| | | |
|:---:|:---:|:---:|
| (a) IMDB-M | (b) PROTEINS | (c) COLLAB |

*Figure 2.* Adversarial examples found by the proposed method. Red edges denote deleted edges from the original samples and green edges indicate those added.

*Table 1.* Validation accuracy of the GCN victim models on the TU datasets before (*clean*) and after various attack methods. Results shown in mean $\pm$ 1 standard deviation across 3 trials.

| | IMDB-M | PROTEINS | COLLAB |
|---|---|---|---|
| *Clean* | $50.53_{\pm 1.4}$ | $71.73_{\pm 2.6}$ | $79.73_{\pm 2.1}$ |
| Random | $47.43_{\pm 1.2}$ | $19.46_{\pm 1.7}$ | $76.41_{\pm 6.2}$ |
| Genetic (Dai et al., 2018) | $47.82_{\pm 1.5}$ | $14.88_{\pm 1.7}$ | $58.61_{\pm 7.9}$ |
| Gradient-based[†] | $\mathbf{39.31_{\pm 2.2}}$ | $50.60_{\pm 4.5}$ | $36.67_{\pm 1.2}$ |
| GRABNEL (ours) | $45.23_{\pm 0.2}$ | $\mathbf{10.82_{\pm 2.5}}$ | $\mathbf{35.38_{\pm 9.3}}$ |

[†]: White-box method

We conduct experiments on three common TU datasets (Morris et al., 2020b) In all cases, we define the attack budget $\Delta$ in terms of the maximum *structural perturbation ratio* $r$ where $\Delta \leq rN^2$ and we set $r = 0.03$. We similarly link the maximum numbers of queries $B$ allowed for individual graphs to their sizes as $B = 50\Delta$, thereby giving larger graphs and thus potentially more difficult instances higher attack and query budgets similar to the conventional image adversarial attack literature (Ru et al., 2019). We cap the maximum number of queries to be $2 \times 10^4$ on any graph. We compare against random search, genetic algorithm originally introduced in Dai et al. (2018) and an additional simple gradient-based method which greedily adds or delete edges based on the magnitude computed input gradient similar to the gradient based method described in Dai et al. (2018) (note that this method is *white-box* as access to parameter weights and gradients is required). Furthermore, noting that the original implementation of RL-S2V, the primary algorithm in Dai et al. (2018), primarily focus on a S2V-based victim model (Dai et al., 2016), we also compare GRAB-NEL against it in the same dataset considered in Dai et al. (2018) in App.C.

We use a Graph Convolutional Network (GCN) (Kipf and Welling, 2017) victim model, and show the classification performance of both victim models before and after attacks using various methods in Table 1: The results show that

GRABNEL typically leads to the largest degradation in victim predictions in all tasks, often performing on par or better than Gradient, a white-box method. Further, GRABNEL typically outperforms in a larger extent for the larger graphs (e.g. COLLAB) on which the benefit of the sequential selection of edge perturbation is more significant.

## 5. Attack Analysis

In this section we provide a qualitative analysis on the common interpretable patterns behind the adversarial samples found, which provides further insights into the robustness of graph classification models against structural attacks. We believe such analysis is especially valuable, as it may facilitate the development of even more effective attack methods, and may provide insights that could be useful for identification of real-life vulnerabilities for more effective defence. We show some examples of the adversarial samples in Fig. 2, and we summarise some key findings below.

*1. Adversarial edges tend to cluster closely together*: We observe that the distribution of the adversarial edges in a graph is highly uneven, with many adversarial edges often sharing common end-nodes or having small spatial distance to each other. This is empirically consistent with recent theoretical findings on the stability of spectral graph filters in Kenlay et al. (2021).

*2. Adversarial edges often attempt to destroy or modify community structures*: for example, the original graphs in the IMDB-M dataset can be seen to have community structure, and the attack tends to flip the edges *between* the communities, and thereby destroying the structure by either merging communities or deleting edges within a cluster. With similar observations also present in, for example, PROTEINS dataset, this may suggest that the models may be fragile to modification of the community structure.

## 6. Conclusion

This work proposes a novel and flexible black-box method to attack graph classifiers using Bayesian optimisation. We demonstrate the effectiveness and query efficiency of the method empirically. Unlike many existing works, we qualitatively analyse the adversarial examples generated. We believe such analysis is important to the understanding of adversarial robustness of graph-based learning models. In future work, we will analyse adversarial patterns quantitatively and consider their effectiveness as a prior to guide the edge selection step of our method. A current limitation is that our model is specific to graph classification. It is straightforward to apply our method to other graph models, and We believe it is also possible to adapt to node classification tasks by suitably modifying the loss function.

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

# Appendix

## A. Optimisation of acquisition function

At each BO iteration, there is need for the inner-loop optimisation of the acquisition function $\alpha(\cdot)$ to select the next point(s) to query $f_\theta$ (we use expected improvement (EI) (Jones et al., 1998) as the default acquisition function). While sample efficiency is typically not an issue in optimising $\alpha$, the problem nonetheless entails the graph combinatorial space on which common tools like gradient-based optimisers cannot be used. We instead optimise $\alpha$ via the genetic algorithm (GA), which has recently been shown to be competitive in BO acquisition optimisation (Cowen-Rivers et al., 2020) and does not require a continuous search space. We outline the ingredients of GA used below:

- *Initialisation:* In our case, we are not totally ignorant about the search space as we might have already queried and observed $f_\theta$ with a few different perturbed graphs $\mathcal{G}'$. A possible smoothness assumption on the search space, for example, would be that if a $\mathcal{G}'$ with an edge $(u, v)$ flipped from $G$ led to a large $\mathcal{L}_{\text{attack}}$, then another $\mathcal{G}'$ with $(u, s), s \notin \{u, v\}$ flipped is more likely to do so too. To reflect this, we fill the initial population by *mutating* the top-$k$ queried $\mathcal{G}'$s leading to the largest $\mathcal{L}_{\text{attack}}$ seen so far in the current stage, where for $\mathcal{G}'$ with $(u, v)$ flipped from the base graph we 1) randomly choose an end node ($u$ or $v$) and 2) change that node to another node in the graph except $u$ or $v$ such that the perturbed edges in all children shares one common end node with the parent.
- *Evolution*: After the initial population is built, we follow the standard evolution routine by evaluating the acquisition function value for each member as its *fitness*, selecting the top-$k$ performing members as the breeding population and repeating the mutation procedure in initialisation for a fixed number of rounds. At termination, we simply query $f_\theta$ with the graph(s) seen so far (i.e. computing the loss in Fig. 1) with the largest acquisition function value(s) seen during the GA procedure.

## B. WL feature extractor

In this section we describe `WLFeatureExtract` in Algorithm 1 in greater detail. The module takes in both the input graph itself and the set of all input graphs (including itself), as the second argument is to construct a collection of all WL features seen in any of the input graphs and controls the dimensionality of the output feature vector so that the entries in feature vectors of different input graphs represent the same WL feature. For an illustrated example of the procedure, the readers are referred to Fig. 3.

## C. Comparison with RL-S2V

We compare GRABNEL with RL-S2V on the graph classification dataset described in Dai et al. (2018). Each input graph is made of 1, 2 or 3 connected components. Each connected component is generated using the Erdős–Rényi random graph model (additional edges are added if the generated graph is disconnected). The label node features are set to a scalar value of 1 and the corresponding graph label is the number of connected components. The authors consider three variants of this dataset using different graph sizes, we consider the variant with the smallest graphs ($15-20$ nodes. The victim model, as well as the surrogate model used to compute Q-values in RLS2V is structure2vec (Dai et al., 2016). This embedding has a hyper-parameter determining the depth of the computational graph. We fix both to be the the smallest model considered in Dai et al. (2018). These choices were made to keep the computational budget to a minimum.

To adapt to the settings in Dai et al. (2018), we only allow one edge edit (addition/deletion), and for GRABNEL we allow up to 100 queries to the victim model per sample in the validation set. For Random baseline, we instead allow up to 400 queries. Similar to Dai et al. (2018), we enforce the constraint such that any edge edit must not result in a change of the number of disconnected components (i.e. the label) and any such edit proposed is rejected before querying the victim model. We show the results in Fig. 5, and we similarly visualise some of the adversarial samples found by GRABNEL in Fig. 4. The final performance of RL-S2V is similar to that reported in Dai et al. (2018), whereas we find that random perturbation is actually a very strong baseline if we give it sufficient query budget[2]. Again, we find that GRABNEL outperforms the baselines, offering orders-of-magnitude speedup compared to RL-S2V, with the main reasons being 1) GRABNEL is designed to be sample-efficient, and 2) GRABNEL does not require a separate training set *within* the validation to train a policy like what RL-S2V does. Fig. 4 shows that the edge addition is more common than deletion in the adversarial examples in this particular case, and often the attack agent forms *ring structures*. Such structures are rather uncommon in the original graphs generated from the Erdos-Renyi generator, and are thus might not be familiar to the classifier during training. This might explain why the victim model seems particularly vulnerable to such attacks.

---

[2]The random baseline reported in Dai et al. (2018) is obtained by only querying victim model with a randomly perturbed graph *once*.

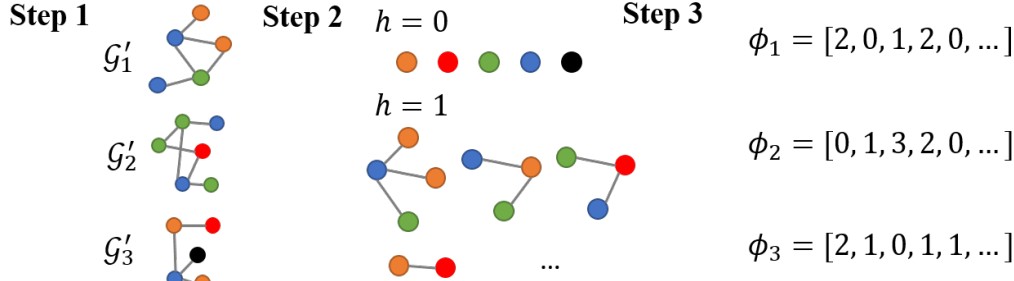

*Figure 3.* Illustration of the WL extractor. Consider an example of an input of three graphs to the extractor $\{\mathcal{G}_1', \mathcal{G}_2', \mathcal{G}_3'\}$ with colors representing the different (discrete) node labels and we would like to compute `WLFeatureExtract`$(\{\mathcal{G}_i', \{\mathcal{G}_1', \mathcal{G}_2', \mathcal{G}_3'\}) \, \forall i$ (**Step 1**). The extractor module takes in 2 arguments, as the second argument consisting of the set of all input graphs is used to generate a collection of all possible Weisfeiler-Lehman features seen in *all input graphs* (**Step 2**), up to $H \in \mathbb{N}^+$ where $H$ is the number of WL iterations specified. This step involves computing the Weisfeiler-Lehman embedding on all of the input graphs using the routine introduced in Shervashidze et al. (2011). The extractor finally counts the number of each features present from Step 2 and outputs the feature vector (**Step 3**; only $h = 0$ part of the feature vector is shown in the figure – note that $\mathcal{G}_1'$ has 2 orange nodes, 2 blue nodes and 1 green nodes which yields the corresponding feature vector $\phi_1$). Note that if a particular feature present in the entire set of input graphs is not present in a particular graph, the entry is filled with zero. *The graphs here are for illustration only; in our task each input graph is only one edit distance different from the base graph $\mathcal{G}_0$.*

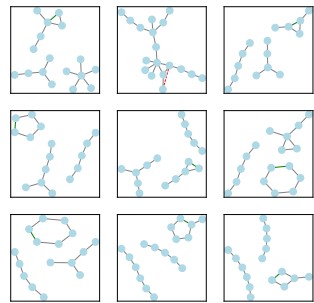

*Figure 4.* Adversarial examples found by the proposed method on the ER graphs with S2V being the victim model. Similar to Fig. 2, Red edges denote deleted edges from the original samples and green edges indicate those added.

## D. Implementation Details

**Datasets**  We provide some key descriptive statistics of the TU datasets (Morris et al., 2020b) in Table 2. All TU datasets may be downloaded at https://chrsmrrs.github.io/datasets/docs/datasets/.

**Computing Environment**  We conduct all experiments on a server with an Intel Xeon CPU with 256GB of RAM.

*Table 2.* Key statistics of the TU datasets used.

| Dataset | #graphs | #labels | Avg #nodes | Avg #edges |
|---------|---------|---------|------------|------------|
| IMDB-M | 1500 | 3 | 13.0 | 65.9 |
| PROTEINS | 1113 | 2 | 39.1 | 72.8 |
| COLLAB | 5000 | 3 | 74.5 | 2457.8 |

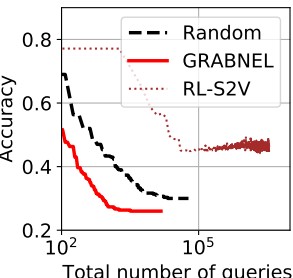

*Figure 5.* Validation accuracy vs *total* number of queries to the victim model. RL-S2V requires significantly more victim model queries, as it attempts to learn an attack policy by repeatedly querying a subset of the validation set which is used for policy training.

**Victim models**  We use the popular graph convolutional network (GCN) (Kipf and Welling, 2017) as the victim model. The graph convolution layers work by aggregating information across the graph edges and then updating combined node features to output new node features. Multiple layers of graph convolution are used. A readout layer transforms the final node embeddings into a fixed-sized graph embedding which can then be fed through a linear layer and a softmax activation function to provide predicted probabilities for each class.

The GCN graph convolutions take the form

$$\mathbf{X}^{(h)} = \sigma(\tilde{\mathbf{D}}^{-1/2}\tilde{\mathbf{A}}\tilde{\mathbf{D}}^{-1/2}\mathbf{X}^{(h-1)}\mathbf{\Theta}^{(h)})$$

**GRABNEL**  GRABNEL, which uses the WL feature extractor, involves a number of hyperparameters: the WL procedure is parameterised by a single hyperparameter $H$,

which specifies the number of Weisfiler-Lehman iterations to perform. While it is possible for $H$ to be selected automatically via, for example, maximising the log-marginal likelihood of the surrogate model (e.g. (Ru et al., 2020)), in our case we find fixing $H = 1$ to be performing well. For the sparse Bayesian linear regression model used, we always normalise the input data into hypercubes $[0, 1]^d$ and standardise the target by deducting its mean and dividing by standard deviation. We optimise the marginal log-likelihood via a simple gradient optimiser and we set the maximum number of iterations to be 300. As described in Sec. 2, we need to specify a Gamma prior over over $\{\lambda_i\}$ and we use shape parameter and inverse scale parameters of $1 \times 10^{-6}$. For the acquisition optimisation, we set the maximum number of evaluation of the acquisition function to be 1000: we initialise with 100 randomly sampled perturbed graphs, each of which is generated from flipping one pair of randomly selected end nodes from the base graph. To generate the initial population, we fill generate 100 candidates by mutating from the top-3 queried graphs that previously led to the largest attack loss (if we have not yet queried any graphs, we simply sample 100 randomly perturbed graphs). We then evolve the population 10 times, with each evolution cycle involving mutating the current population to generate offspring and popping the oldest members in the population. Finally, we select the top 5 unique candidates seen during the evolution process that have the highest acquisition function value (we use the expected improvement (EI) acquisition function) to query the victim model.