# OpenReview forum: "Attacking Graph Classification via Bayesian Optimisation"
_ICML.cc/2021/Workshop/AML — ICML 2021 Workshop AML Poster_

### Official Review · Reviewer_Stiv · 2021-06-20
**An easy method for black-box graph level classification attack**

**Rating:** Accept
**Confidence:** 4

**Review:**

The paper proposes to use BO to attack graph-level classifier in a black-box fashion. The authors define a WL-based feature extractor followed by a Bayesian linear classifier as the surrogate model for BO and introduce a sequential procedure to apply perturbations. Empirical studies demonstrate the effectiveness and query efficiency of the method.

The part I like most about this paper is the inspection of the common interpretable patterns behind the adversarial samples
found, which provides further insights into the robustness of graph classification models against structural attacks.

Regarding the shortcomings, I think the methods developed in the paper are fairly simple and make an analogy to some existing works on compositional optimization via BO. The authors should rethink and clarify their technical contributions.

By the way,  the current surrogate model is not so flexible, can it approach some modern GNNs in performance?
Further,  the Bayesian linear classifier may not provide meaningful extrapolations for non-linear inputs. Can you use stronger Bayesian models like GPs or BNNs?

---

### Decision · Program_Chairs · 2021-06-21

**Decision:**

Accept (Poster)

**Comment:**

This paper proposed to use Bayesian Optimization to attack graph classifiers in a black-box fashion. The authors can further address the reviewer's comments.